# Quantum Honeypots

**DOI:** 10.3390/e25101461

**Published:** 2023-10-18

**Authors:** Naya Nagy, Marius Nagy, Ghadeer Alazman, Zahra Hawaidi, Saja Mustafa Alsulaibikh, Layla Alabbad, Sadeem Alfaleh, Areej Aljuaid

**Affiliations:** 1Department of Networks and Communication, College of Computer Science and Information Technology, Imam Abdulrahman Bin Faisal University, Dammam 31441, Saudi Arabia; gralazman@iau.edu.sa (G.A.); zahra.hawaidi@outlook.com (Z.H.); sj.mustafa.ah@gmail.com (S.M.A.); layla997alabbad@gmail.com (L.A.); sadeem.faleh@hotmail.com (S.A.); areejaljuaid@outlook.com (A.A.); 2College of Computer Engineering and Science, Prince Mohammad Bin Fahd University, Al-Khobar 31952, Saudi Arabia; mnagy@pmu.edu.sa

**Keywords:** honeypot, post-quantum security, quantum security, quantum networks

## Abstract

Quantum computation offers unique properties that cannot be paralleled by conventional computers. In particular, *reading* qubits may change their state and thus signal the presence of an intruder. This paper develops a proof-of-concept for a quantum honeypot that allows the detection of intruders on reading. The idea is to place quantum sentinels within all resources offered within the honeypot. Additional to classical honeypots, honeypots with quantum sentinels can trace the reading activity of the intruder within any resource. Sentinels can be set to be either visible and accessible to the intruder or hidden and unknown to intruders. Catching the intruder using quantum sentinels has a low theoretical probability per sentinel, but the probability can be increased arbitrarily higher by adding more sentinels. The main contributions of this paper are that the monitoring of the intruder can be carried out at the level of the information unit, such as the bit, and quantum monitoring activity is fully hidden from the intruder. Practical experiments, as performed in this research, show that the error rate of quantum computers has to be considerably reduced before implementations of this concept are feasible.

## 1. Introduction

Honeypots are software-based security devices that operate within the general effort of protecting computing systems: servers, databases, networks, or, more generally, organizations. A honeypot is [1] intentionally constructed to be attacked, explored, and compromised. It is frequently used for detecting and dispersing unauthorized activities. Furthermore, its primary functionality is to investigate the conduct of attackers and to experience and pinpoint specific unknown attacks. The definition may vary between authors, a close to unifying definition places the value of a honeypot in its characteristics of being open to be inspected, attacked, and ultimately compromised [2]. The concept of honeypots moves the defence strategy from a passive paradigm to a proactive paradigm. Rather than building a strong defence for the sensitive system and waiting for an attacker to try out various attacks, the honeypot approach walks a totally different path by creating an alternate/fake environment that is offered to attackers. Attackers unknowingly try to exploit the fake environment and are mislead to fake resources. Thus, honeypots are dedicated to attracting hackers by presenting services and open ports that are potentially vulnerable. The purpose is to monitor and analyze the activities of hackers and intruders in a way in which they do not know that they are being observed. Further, current attack methods and trends can be classified and studied in order to find the appropriate protection.

Quantum computation already has a mature theoretical background [3] with a fast emerging practical technology. In terms of theoretical results, quantum computational results have been achieved by algorithms with asymptotic speed-ups over classical counterparts: Grover’s search algorithm [4], Shor’s factorization algorithm [5], and others. Nevertheless, arguably the most successful branch of quantum computation is quantum cryptography, with the promise of unconditionally secure communication protocols [6]. Quantum security protocols have been designed for key distribution [7], delayed secure decisions, and zero knowledge protocols. Note that, in all these security primitives, the quantum protocols plays a passive role in the protection of a system. Our paper, by contrast, is the first to propose an *active* quantum protocol of the type of a honeypot. It will be shown that, by employing quantum networks and quantum communication systems, the honeypot can be enhanced with additional monitoring capabilities, while fully hiding its presence.

The idea in this paper is to use quantum computation techniques to better hide the fact that the services offered to the attackers are fake, as well as to monitor the intruder without being detected. This is achieved through quantum sentinels. Quantum sentinels is a new concept proposed for the first time in this paper. We define two types of sentinels: *positional sentinels* with recognizable positions and *hidden sentinels*, which are not detectable from the outside. In the first case, the intruder is able to directly read and otherwise act on the *positional* sentinel. The position of the sentinel, within the array of qubits, may be secret or public, yet by direct reading, its existence is still visible to the outside world. In the second case, the *hidden* sentinel cannot be seen or read from outside and is thus accessible only to the honeypot system. The hidden sentinel relies on the quantum Fourier transform to connect to the luring datum qubit. The quantum sentinels flag the presence of an intruder when the intruder *reads* some information. The information may be part of a file, an address, the content of memory or a hard disk sector.

In terms of technological readiness for the commercial implementation of a quantum honeypot scheme, consider that a mere decade ago, quantum computers were technologically questionable [8], whereas the situation now shows that quantum supremacy has been affirmed to be achieved by several academic and commercial sources [9,10].

A honeypot may present itself as an entire system, such as a node in the internet, that looks as if it contains useful information and data, but, in reality, is meant only to lure unlawful activities. The interface to the outside world is there, but in the background there is no useful application. Any honeypot consists of two essential elements: decoy and captor. The decoy lures the attacker by offering information system resources, whether physical or virtual. The captor is the part of the system that actually inspects and records the activity of the intruder. It acts in detecting the intruder, responding to the requests, and profiling the attacker. Our proposed quantum sentinels are in the category of the Captor. Additionally to the classical captor, these sentinels can be fully hidden.

There are three categories of honeypots depending on the information system resources that are provided to the attacker: low-interaction honeypot, medium-interaction honeypot, and high-interaction honeypot [11]. As the names suggest, the three levels allow for increasing penetration capability. High-interaction honeypots define an entire operating system for tampering. Quantum sentinels can be added to all elements of a functional operating system, making them suitable for high-interaction honeypots. Thus, with quantum sentinels, the entire activity of the attacker can be monitored to a more detailed level. In fact any reading of information can be monitored both in terms of the actual reading action as well as the time of the action.

This paper builds on the idea that *any* activity within the honeypot is categorized as malicious [11]. With quantum sentinels added to honeypots, the extent of malicious activity can be detailed to the next level, where hackers may be caught on any particular bit they access for reading. Additionally, the honeypot exhibits a better hiding effect. Note that the purpose of the design in this paper is the recognition of the intruder’s behavior and that it does not deal directly with allowing the intruder to compromise the system.

The rest of the paper is organized as follows. Section 2 describes the quantum properties used in the honeypot algorithms as well as the quantum network setting. Section 3 presents the honeypot quantum algorithms that capture the activity of the intruder on the simple reading of any storage medium. The difference between *positional sentinels* and *hidden sentinels* is also described here. Section 4 shows the behavior of the algorithms as implemented on a real quantum computer, IBMQX, as a proof of concept of how quantum sentinels work. Section 5 concludes the paper.

## 2. Quantum Properties

Discussions about quantum computation revolve not only around quantum supremacy [12], but also around quantum network communication. Quantum network communication algorithms are studied for various problems. García-Cobo [13] defines a quantum algorithm for key distribution within a large quantum network. The network experiments have been done with simulations over a known geographical territory in Castilla using quantum repeaters to propagate quantum signals.

In our setting, we suppose to have a network with quantum connections and the devices connected to the network are also able to do quantum computations on qubits, at least in part of the memory.

The quantum honeypot connects to the outside worlds through quantum connection. Users, such as fake users and hackers, communicate with the honeypot via quantum channels. Quantum channels allow the bidirectional transmission of messages. In order to put no limitation on the amount of communication, we consider these messages to be arbitrarily large. Note that this assumption is common in quantum algorithms, such as quantum key distribution [7,14], where the analysis of the algorithm allows the size of the message transmitted from one partner to the other to be arbitrarily long.

### 2.1. Measurement and State Collapse

A qubit in Dirac’s notation [3] is a superposition of the base vectors |0〉 and |1〉.
q=α|0〉+β|1〉,
where the coefficients α,β and the amplitudes are complex numbers and the vector is of a unitary norm, i.e., $\sqrt{{\left|\alpha \right|}^{2}+{\left|\beta \right|}^{2}}=1$. There are two specific balanced super-positions, namely $|+\rangle =\frac{1}{\sqrt{2}}(|0\rangle +\left|1\rangle \right)$ and $|-\rangle =\frac{1}{\sqrt{2}}(|0\rangle -\left|1\rangle \right)$. They will play a role in the algorithms below.

When qubits are measured, they are measured on some basis. The simplest measurement base is the computational base, with the base vectors |0〉 and $|1\rangle =\frac{1}{\sqrt{2}}(|0\rangle -\left|1\rangle \right)$. This base is not unique. Another common measurement base is the Hadamard base, with base vectors $|+\rangle =H|0\rangle =\frac{1}{\sqrt{2}}(|0\rangle +\left|1\rangle \right)$ and |−〉=H|1〉. Both these bases are ortho-normal bases. In any case, when an arbitrary qubit is measured, the state of the qubit collapses to one of the base vectors. Thus, an arbitrary qubit q=α|0〉+β|1〉 can be measured in multiple measurement bases. When *q* is measured in the computational basis, it collapses either to |0〉 with probability $\alpha^{2}$, or to |1〉 with probability $\beta^{2}$. Again, when *q* is measured in the Hadamard basis, it collapses either to |+〉 or to |−〉. The probabilities of collapse can be seen from rewriting the qubit as $q=\alpha |0\rangle +\beta |1\rangle =\frac{\alpha +\beta }{\sqrt{2}}\frac{1}{\sqrt{2}}(|0\rangle +\left|1\rangle \right)+\frac{\alpha -\beta }{\sqrt{2}}\frac{1}{\sqrt{2}}(|0\rangle -\left|1\rangle \right)$. *q* is measured as |+〉 with probability $\frac{{\left(\alpha +\beta \right)}^{2}}{2}$ and as |−〉 with probability $\frac{{\left(\alpha -\beta \right)}^{2}}{2}$.

### 2.2. Qubits and Quantum Gates

Quantum gates can apply on one or more qubits. The condition on quantum gates is that they be reversible. Therefore, all quantum gates have the same number of inputs and outputs. Here is a list of the gates used in our algorithms.

AThe **Hadamard gate**

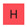
 rotates the states |0〉 and |1〉 into |+〉 and |−〉, respectively. It contributes towards a universal gate set on a quantum computer as the single quantum gate needed in addition to a universal gate set for classical computation. The Hadamard gate is useful for creating balanced superpositions. The reverse is also true, namely that a Hadamard gate applied to a balanced superposition brings the qubit to the respective base state. The Hadamard gate is its own inverse.BThe **NOT gate**

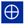
 is also known as the Pauli-X gate and, in this form, is also applied on a single qubit. The |0⟩ state flips to |1⟩ and vice versa. As shown below, it is represented by the Pauli matrix:
$$X=\left[\begin{array}{cc} 1 & 0\\ 0 & 1 \end{array}\right]$$CThe **Controlled Phase Shift gate** is a two qubit gate, built from a simple phase shift gate.The simple phase shift gate ($R_{z}$) operates on a single qubit. It rotates the qubit around the z axis of the Bloch sphere [3]. Thus, the gate changes the phase and the angle of the |0〉 and |1〉, but not the respective percentages of the two within the superposition.The controlled phase shift gate, $CR_{z}$, has an additional control qubit 
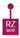
. This gate is a two qubit gate. The control modifier determines whether the shift is applied or not on the original data-qubit. If the control is |1〉, the gate is active, and it is inactive for a control equal to |0〉.

In the following algorithms, the phase shift gate always has the rotation angle of either $\frac{\pi }{2}$, or its opposite $-\frac{\pi }{2}$. Consider a few applications of the transformation, as they appear in this study. The rotation of $\pm \frac{\pi }{2}$ on the computational base vectors has no effect, $R_{z}^{\pm \frac{\pi }{2}}|0\rangle =|0\rangle$ and $R_{z}^{\pm \frac{\pi }{2}}|1\rangle =|1\rangle$. On the Hadamard base vectors, the $\frac{\pi }{2}$ rotation changes the phase, $R_{z}^{\frac{\pi }{2}}|+\rangle =\frac{1}{\sqrt{2}}(|0\rangle +i|1\rangle$ and $R_{z}^{\frac{\pi }{2}}|-\rangle =\frac{1}{\sqrt{2}}(|0\rangle -i|1\rangle$. A double application of $R_{z}^{\frac{\pi }{2}}$ moves from one Hadamard base vector to the other, $R_{z}^{\frac{\pi }{2}}R_{z}^{\frac{\pi }{2}}|+\rangle =|-\rangle$ and $R_{z}^{\frac{\pi }{2}}R_{z}^{\frac{\pi }{2}}|-\rangle =|+\rangle$. In the case of a rotation with $-\frac{\pi }{2}$, the sign before the imaginary term *i* is reversed.

## 3. Quantum Sentinels

Sentinels in computer science are entities, such as variables, that block the access of a program to a certain area or that flag a state of error/emergency within a program. In the case of a quantum sentinel, as described here, we use the latter situation. Quantum sentinels are quantum entities, primarily qubits, that flag a state of emergency. Within a honeypot, a quantum sentinel marks the presence of an intruder.

The physical principle on which quantum sentinels function is the collapse of superposition at measurement. A collapse of superposition can then potentially be detected by a legitimate checking system.

We define two types of quantum sentinels:**Positional sentinels**, which are visible to the intruder, though their quantum state is unknown.**Hidden sentinels**, which are hidden from the user. In this case, both the quantum state and the operation of the sentinel remain unknown to the intruder.

Quantum sentinels capture the action of reading. As such, they can be placed in any part of a computer system where the reading of information is possible. This can be viewed as practical devices that carry information: hard disks, random access memory, network card information, video card memory, external memory-carrying devices, etc. Alternatively, quantum sentinels can be viewed as logically positioned in places holding key information: operating system settings, boot sectors of hard disks, meta data storage files, log and important history files, configuration files, environment variable settings, keyword and password locations, sensitive information locations, or crucial data paths. The two types of quantum sentinels presented here have different behaviors and, therefore, can be employed with somehow different purposes. The positional sentinel marks an important information carrier, such as a sensitive parameter setting for an application or an operating system. In this case, the sentinel may be visible, and the intruder may know of the presence of sentinels, while being unable to avoid reading them. The hidden sentinel has a more insidious capability of remaining unnoticed for the entire activity of the intruder. In this case, a longer observation of the intruder is possible; it is possible to trace the intruder in terms of actions, using their target and timeline to get the entire action plan of the intrusion. Hidden sentinels are more costly, as they involve a small circuit for each quantum sentinel, they should be employed more sparingly according to needs, whereas positional sentinels are less involved.

The idea of quantum sentinels comes from the fact that an unknown quantum state, when read, collapses along the measurement basis. Thus, using two measurement bases, a qubit read by an intruder in the wrong basis changes its original state and this state change can be detected by the honeypot server, with a certain probability. This property has been used in public key distribution [7]. Thus, consider the two measurement bases:The Computational Basis, with the base vectors |0〉 and |1〉The Hadamard Basis, with the base vectors |+〉 and |−〉

Suppose that the only allowed states of a qubit are the four base vectors above: |0〉, |1〉, H|0〉, and |H|1〉.

The sentinel capacity comes from the qubit’s unknown state to the intruder. The interplay between computational and Hadamard bases makes the qubit’s state vulnerable to changes when accessed by an unknowing intruder.

### 3.1. Positional Sentinels

Positional sentinels refer to sentinels that are controlled by the server and that the client can see.

A positional quantum sentinel is a qubit in one of the four states: |0〉, |1〉, |+〉, or |−〉. When read, a sentinel qubit may keep its exact state or may change the state to align with the reading basis, computational or Hadamard.

Depending on the value of the qubit, reading the qubit may or may not collapse the qubit to another value and thus change, or not change, its state. A qubit of state |0〉 or |1〉 does not change when measured in the computational basis, but when measured in the Hadamard basis, it collapses to |+〉 or |−〉. The exact reverse happens for a qubit in state |+〉 or |−〉; when measured in the Hadamard basis, the quantum state remains unchanged, and when measured in the computational basis, the qubit collapses to |0〉 or |1〉. Thus, a sentinel has to be measured in the correct basis in order to remain unchanged. Consider that a valid user is knowledgeable about the inherent basis of the sentinel qubit.

This means that an intruder, not knowing the basis of the sentinel, risks changing its state by inadvertent reading. Table 1 synthesizes all the possibilities of a sentinel value versus the reading options of the legitimate user and the intruder. The conclusion is that a sentinel catches an intruder with a probability of $\frac{1}{4}$. To improve the probability overall, a simple addition of sentinels in desired positions increases the probability to any desired value. One positional sentinel remains undetected with probability $1-\frac{1}{4}=\frac{3}{4}$. Thus, the detection rate for *m* positional sentinels is
$$p_{positional}^{m}=1-{\left(\frac{3}{4}\right)}^{m}.$$

To exemplify this growth, eight sentinel qubits catch an intruder with probability $1-{\left(\frac{75}{100}\right)}^{8}=89.9\%$ and the probability grows exponentially with the number of sentinels.

The server can check the state of a sentinel by reading it in the correct basis. The drawback of positional sentinels is that they are exposed to the intruder. The secrecy of the position of the sentinels may be part of the honeypot concept, but the sentinel itself is part of the information read by the intruder. Thus, as the value of the sentinel depends on the reading basis, its value may affect the meaning of the information read by the intruder and, thus, reveal the presence of the sentinel itself. Nevertheless, this is not all the capability of quantum sentinels, as sentinels themselves can remain fully hidden from the intruder.

### 3.2. Hidden Sentinels

Hidden sentinels are sentinels controlled by the server that the intruder cannot see. They are physically not addressable by the user or intruder. A hidden quantum sentinel is a qubit that is never exposed to the user, but is connected through entanglement to a datum-qubit that is available to the intruder to read. The main idea is that, when an intruder reads the available datum-qubit in a wrong basis, the sentinel’s state changes to a value consistent with the measurement of the datum-qubit and this change is detectable by the honeypot server.

The circuit involved for every hidden sentinel is simple, using two gates: a Hadamard gate and a phase shift gate. It is a portion of the quantum Fourier transform.

The Quantum Fourier Transform [15] allows the phase of a qubit to be changed, that is, the qubit is rotated around the Oz axis. The value of the rotation is given by a series of control qubits in such a way that the impact of each successive control qubit is half the angle rotation effect of the previous one. In our case, we are interested in only one control qubit, see Figure 1. The top qubit is the datum-qubit, which acts as the control qubit to the gates. This is also the qubit that the intruder acts on. The phase shift gate, in purple, acts on the second qubit. The second qubit is the sentinel. If the datum-qubit is one, then it has an effect on the sentinel qubit, as it injects a rotation of the phase with $\frac{\pi }{2}$. The sentinel undergoes two such gates in reverse. The first gate prepares the sentinel before the honeypot is offered to attack. The second gate is used for checking the state of the sentinel. Note that each time a sentinel is checked, the sentinel is destroyed. This means that, to re-activate a hidden sentinel, the sentinel has to undergo preparation again.

The behavior of the hidden sentinel is also based on the intruder not knowing the correct reading basis of the data qubit. If lucky, the intruder will not be caught. If unlucky, the intruder will be caught with a chance of $\frac{1}{4}$. As the intruder may or may not be lucky with equal probability, the overall theoretical probability to catch an intruder with a hidden sentinel is $\frac{1}{8}$. This is half of the probability of a positional sentinel. Thus, there is a drawback of using hidden sentinels, in that the probability is lower.

The following formulas describe the situations in detail. The ensemble of two qubits is always written with the *datum first* and the *sentinel second*, $|D\rangle |S\rangle =q_{1}q_{0}$. The preparation of the sentinel, as shown in Figure 1, can be described by the transformation $R_{z}^{\frac{\pi }{2}}\left(I_{2}⊗H\right)$. The server’s checking of the sentinel is similar, namely $\left(I_{2}⊗H\right)R_{z}^{-\frac{\pi }{2}}$. Additionally, if the intruder is unlucky, the data qubit may undergo a Hadamard transformation, as imposed by the intruder; this is the transformation $H⊗I_{2}$.

Consider **the data qubit to be** |0〉. In this case, the initial state of the system is |D〉|S〉=|0〉|0〉. If the intruder is lucky and uses the computational basis himself, the following transformation happens to the system.
(1)$$\begin{array}{ccc} result-lucky & = & \left(I_{2}⊗H\right)R_{z}^{-\frac{\pi }{2}}\left(I_{2}⊗I_{2}\right)R_{z}^{\frac{\pi }{2}}\left(I_{2}⊗H\right)\;|0\rangle |0\rangle \\ & = & \left(I_{2}⊗H\right)R_{z}^{-\frac{\pi }{2}}\left(I_{2}⊗I_{2}\right)R_{z}^{\frac{\pi }{2}}\;|0\rangle |+\rangle \\ & = & \left(I_{2}⊗H\right)R_{z}^{-\frac{\pi }{2}}\;|0\rangle |+\rangle \\ & = & (I_{2}⊗H)\;|0\rangle |+\rangle =|0\rangle |0\rangle \end{array}$$

The operation in red represents the action of the intruder, which, in this case, is the identity transformation that is no action at all. This is because this measurement is aligned with the state of the qubit and does not change the state of the system. The sentinel is found in its original value |0〉. The server checker concludes that no intrusion happened.

Now consider that the intruder makes the mistake and reads in the Hadamard basis.
(2)$$\begin{array}{ccc} result-unlucky & = & \left(I_{2}⊗H\right)R_{z}^{-\frac{\pi }{2}}\left(H⊗I_{2}\right)R_{z}^{\frac{\pi }{2}}\left(I_{2}⊗H\right)\;|0\rangle |0\rangle \\ & = & \left(I_{2}⊗H\right)R_{z}^{-\frac{\pi }{2}}\left(H⊗I_{2}\right)\;|0\rangle |+\rangle \\ & = & \left(I_{2}⊗H\right)R_{z}^{-\frac{\pi }{2}}\;|+\rangle |+\rangle \\ & = & \left(I_{2}⊗H\right)R_{z}^{-\frac{\pi }{2}}\;(\frac{1}{\sqrt{2}}|0\rangle |+\rangle +\frac{1}{\sqrt{2}}\left|1\rangle \right|+\rangle )\\ & = & \left(I_{2}⊗H\right)\;(\frac{1}{\sqrt{2}}|+\rangle |0\rangle +\frac{1}{2}|1\rangle (|0\rangle +i|1\rangle )) \end{array}$$

In this case, when the sentinel is measured by the checker in the computational basis, the theoretical probability to measure |0〉 is $\frac{3}{4}$ and the probability to measure a |1〉 is $\frac{1}{4}$. If the checker sees a 1, this value indicates the presence of the intruder.

Overall, the probability of the intruder being caught by the checker, in this case, is $p=\frac{1}{2}*\frac{1}{4}=\frac{1}{8}$. Though the calculations vary from case to case, the overall result is shown to be the same. When **the data qubit is**|1〉, the formulas are very similar with some difference in the signs.

Consider **the data qubit to be** |+〉. In this case, the initial state of the system is |D〉|S〉=|+〉|0〉. If the intruder is lucky and uses the Hadamard basis of measurement, then the measurement is aligned with the actual qubit value and its state is undisturbed. This means that no change is applied to the data qubit, and the measurement is formally described by $I_{2}⊗I_{2}$. The system is transformed as follows.
(3)$$\begin{array}{ccc} result-lucky & = & \left(I_{2}⊗H\right)R_{z}^{-\frac{\pi }{2}}\left(I_{2}⊗I_{2}\right)R_{z}^{\frac{\pi }{2}}\left(I_{2}⊗H\right)\;|+\rangle |0\rangle \\ & = & \left(I_{2}⊗H\right)R_{z}^{-\frac{\pi }{2}}\left(I_{2}⊗I_{2}\right)R_{z}^{\frac{\pi }{2}}\;|+\rangle |+\rangle \\ & = & \left(I_{2}⊗H\right)R_{z}^{-\frac{\pi }{2}}\left(I_{2}⊗I_{2}\right)R_{z}^{\frac{\pi }{2}}\;(\frac{1}{\sqrt{2}}|0\rangle |+\rangle +\frac{1}{\sqrt{2}}\left|1\rangle |+\rangle \right)\\ & = & \left(I_{2}⊗H\right)R_{z}^{-\frac{\pi }{2}}\left(I_{2}⊗I_{2}\right)\frac{1}{\sqrt{2}}(|0\rangle |+\rangle +|1\rangle \frac{|0\rangle +i|1\rangle }{\sqrt{2}})\\ & = & \left(I_{2}⊗H\right)R_{z}^{-\frac{\pi }{2}}\;\frac{1}{\sqrt{2}}(|0\rangle |+\rangle +|1\rangle \frac{|0\rangle +i|1\rangle }{\sqrt{2}})\\ & = & \left(I_{2}⊗H\right)\;\frac{1}{\sqrt{2}}\left(|0\rangle |+\rangle +|1\rangle |+\rangle \right)=|+\rangle |0\rangle \end{array}$$

The sentinel preserves its original value |0〉 and no intruder can be detected.

Nevertheless, when the intruder mistakenly measures in the computational basis, the sentinel is changed. Note that, in this case, the intruder actually affects the |D〉 qubit and its state is collapsed along the computational basis, which is equivalent to applying a Hadamard gate in the middle.
(4)$$\begin{array}{ccc} result-unlucky & = & \left(I_{2}⊗H\right)R_{z}^{-\frac{\pi }{2}}\left(H⊗I_{2}\right)R_{z}^{\frac{\pi }{2}}\left(I_{2}⊗H\right)\;|+\rangle |0\rangle \\ & = & \left(I_{2}⊗H\right)R_{z}^{-\frac{\pi }{2}}\left(H⊗I_{2}\right)R_{z}^{\frac{\pi }{2}}\;|+\rangle |+\rangle \\ & = & \left(I_{2}⊗H\right)R_{z}^{-\frac{\pi }{2}}\left(H⊗I_{2}\right)R_{z}^{\frac{\pi }{2}}\;(\frac{1}{\sqrt{2}}|0\rangle |+\rangle +\frac{1}{\sqrt{2}}\left|1\rangle |+\rangle \right)\\ & = & \left(I_{2}⊗H\right)R_{z}^{-\frac{\pi }{2}}H⊗I_{2})\frac{1}{\sqrt{2}}(|0\rangle |+\rangle +|1\rangle \frac{|0\rangle +i|1\rangle }{\sqrt{2}})\\ & = & \left(I_{2}⊗H\right)R_{z}^{-\frac{\pi }{2}}\;\frac{1}{\sqrt{2}}(|+\rangle |+\rangle +|-\rangle \frac{|0\rangle +i|1\rangle }{\sqrt{2}})\\ & = & \left(I_{2}⊗H\right)\;\frac{1}{2}(|0\rangle |+\rangle +|1\rangle \frac{|0\rangle +i|1\rangle }{\sqrt{2}}+|0\rangle \frac{|0\rangle +i|1\rangle }{\sqrt{2}}+|1\rangle |+\rangle )\\ & = & \frac{1}{2}(|0\rangle |0\rangle +|1\rangle \frac{|0\rangle -i|1\rangle }{\sqrt{2}}+|0\rangle \frac{|0\rangle -i|1\rangle }{\sqrt{2}}+\left|1\rangle |0\rangle \right) \end{array}$$

From the formula, it can be seen that the probability to get 1 when the checker measures is, again, $\frac{1}{4}$.

Considering that the intruder is lucky or unlucky with equal probability, the checker catches the intruder with a probability of $\frac{1}{8}$. A similar result can be obtained for **a data qubit of**
|−〉.

The overall conclusion is that a hidden sentinel can catch the intruder with a probability of $\frac{1}{8}$. This probability is lower than in the case of positional sentinels, but, again, can be increased arbitrarily higher by adding more sentinels in the area of interest. To evaluate the detection rate of *m* hidden sentinels, note one hidden sentinel remains undetected with probability $1-\frac{1}{8}=\frac{7}{8}$. *m* hidden sentinels remain undetected with probability ${\left(\frac{7}{8}\right)}^{m}$. Thus, the detection rate for *m* hidden sentinels can be computed using the formula
$$p_{hidden}^{m}=1-{\left(\frac{7}{8}\right)}^{m}.$$

To get a perception of how the detection rate changes exponentially with the value *m*, consider that for only m=8 sentinels, the detection rate is already $p_{hidden}^{8}=1-{\left(\frac{7}{8}\right)}^{8}=66\%$.

There is an additional justification on using controlled phase shift gates for setting up hidden sentinels. Note that the data qubit has to explicitly act on the hidden sentinel, whenever touched by a user or intruder. This must be carried out by an explicit two-qubit gate. A simple Bell state type entanglement of the data and the sentinel cannot work here. The reason is that entanglement cannot be used as an information carrier. Otherwise, information could be transmitted instantaneously, rather than, at most, the speed of light, which is the tenet of today’s physics.

It is now the time to check the experiments and how they fit the theoretical calculations.

## 4. Quantum Implementation and Experiments

Consider that the quantum server opens a honeypot that includes quantum sentinels. Thus, the regular services offered by the honeypot are peppered with quantum sentinels, within non-volatile, volatile memory locations or any other bit/qubit arrays. We have defined two types of sentinels, positional and hidden, and they have different characteristics in terms of visibility to the user and efficiency in catching the intruder.

All experiments are implemented using IBM Quantum Experience [16] with a real quantum processor, and both types of sentinels were implemented. The results for each quantum circuit are collected from 1024 runs. The results show a general alignment to the theoretical expectations, except that they are unreliable to a certain degree and sometimes give spurious results.

Experiments were performed for both legal users and intruders. A legal user knows the state of the sentinels and acts accordingly. The intruder takes the best guess, which, in the case of quantum sentinels, is simple random guessing.

### 4.1. Experiments with Positional Sentinels

In the case of positional sentinels, each qubit can play the role of a sentinel. The availability of qubits gives the size of the experiment, namely, to four sentinels. In each experiment, we have two players, the server and the client. The server prepares the honeypot sentinels and the client exploits them. In all figures, the red rectangle pertains to the server and the green rectangle pertains to the client. Measurement is also part of the server activity.

The first type of experiment is that the server generates all possible types of positional sentinels. Recall that there are four types of sentinels according to the four base vectors of the computational and Hadamard measurement bases, |0〉, |1〉, |+〉, and |−〉, in some arbitrary order. Thus, the server side is fixed. It remains for us to define the behavior of the client. We show two directions. The first direction defines a circuit with a legal user, and the second direction contains two experiments where the user is an intruder.

#### 4.1.1. Positional Sentinels with a Legal User

A legal user knows the setting of the sentinels and reads them correctly, see Figure 2. Note that the sentinels have been prepared as $q_{0}=|-\rangle$, $q_{1}=|0\rangle$, $q_{2}=|+\rangle$, and $q_{3}=|1\rangle$. The legal users measures correctly, namely $q_{0}$, $q_{2}$ in the Hadamard Basis, and $q_{1}$, $q_{3}$ in the computational basis.

The results obtained after running the circuit (see Figure 3) show that, indeed, the legal user is correctly classified as such, 93.5% times. Nevertheless, the false negatives are not negligible at 6.5%.

#### 4.1.2. Positional Sentinels with an Intruder as User

The intruder does not know the settings of the sentinels and, therefore, has no other options than to randomly choose the reading bases. The next two experiments show the intruder with two different choices. The randomization options of the client actions were carried out on a randomization tool.

For the first “*intruder*” experiment, the server has prepared the four sentinels as $q_{0}=|0\rangle$, $q_{1}=|+\rangle$, $q_{2}=|1\rangle$, and $q_{3}=|-\rangle$ (see Figure 4). It can be seen in the same figure that the client happened to read the sentinels in the following bases: computational, Hadamard, Hadamard, and Hadamard. Thus, the only sentinel that is wrongly read is qubit $q_{2}$. Here, the server prepared the qubit in the computational basis, but the client used the Hadamard basis for reading instead. We expect the intruder client to be caught with a probability of $\frac{1}{2}$.

The measurement that the server expects from a legal user is 1100. Theoretically, when the server measures 1000, this signals the presence of the intruder. Figure 5 shows the actual measured probabilities. It can be seen that the intruder catching measurement for 1000 is 44.727%, which is significantly different from 50%. The problem is that the quantum computer produces spurious results as well. This is the case for all the values different from 1100 and 1000. Because the server expects exactly 1100 from the legal user, it means that all spurious results contribute to catching the user. Thus, the measured percentage of positively signaling the user becomes 100−50.879%=49.121%, which is very close indeed to the theoretical expectation.

For the second “*intruder*” experiment (see Figure 6), the intruder makes three mistakes, on $q_{0}$, $q_{1}$, and $q_{3}$, respectively. The probabilities of the server to measure the expected 1100 is theoretically ${\frac{1}{2}}^{3}=12.5\%$. All other binary measurements reveal the presence of the intruder, which, again, would theoretically be 100%−12.5%=87.5%.

The practical measurement results, as shown in Figure 7, detect the intruder with 1−11.426%=88.574% probability. Again, we see a slight deviation from the theoretical expectation, but within workable limits.

The conclusion may be that the implementation of positional sentinels seems to be close to feasibility in practical cases. Some false positives and some false negatives have to be contended with.

### 4.2. Hidden Sentinels

In the case of hidden sentinels, the data qubit is considered to be in one of the four basic states, |0〉, |1〉, |+〉, or |−〉. Note that the data qubit does not have an arbitrary value, but has to follow one of these choices. Nevertheless, the user can read it freely, oblivious to the presence or absence of the sentinel, as the sentinel itself is *another* qubit. The intervention of an intruder can be tested by applying the quantum Fourier transform twice: directly and in reverse. The datum qubit serves as the control to the phase shift rotation gates, $CR_{z}$ with rotation $\frac{\pi }{2}$ and $-\frac{\pi }{2}$ and the sentinel qubit is the qubit that the gates act on. The circuit that implements this transform is shown in Figure 8. The datum, $q_{0}$, is set to an initial state and then at the end reset to |0〉. This state is not fixed as described above. The middle of the circuit shows the action of the user, encircled in a golden rectangle. This action will also be variable, depending on the intruder’s choice. The meaning that this circuit offers is that, if the sentinel, $q_{1}$, is measured to the value 0, then the conclusion is that the user is legal and if the value is 1 then the sentinel signals an intruder. As hidden sentinel experiments need two qubits for each experiment, the circuit represents one such sentinel setting.

Figure 8 shows the circuit for a legal user or a lucky intruder, whereas Figure 9 shows the circuit with an unlucky intruder. When run, the circuit with a legal user shows an approximate 10% of false positives, whereas the circuit with the illegal user has a similar deviation from the theoretical expectation. The question remains: how does the error of the quantum computer scale in the case of several sentinels?

#### 4.2.1. Errors on Hidden Sentinels

The problem with available quantum computers today is that their error rate is still prohibitively high. In a real honeypot with quantum sentinels, the number of sentinels should be peppered over all resources, bringing them, in number terms, to a fraction of the entire address space. Nevertheless, the erroneous quantum measurements make this scenario unrealistic, as the following example will show.

Consider a sample circuit with two hidden sentinels (see Figure 10), such that the intruder has been unlucky on both data qubits, that is, the intruder has been consistently unlucky. Qubits $q_{0}$ and $q_{1}$ form the first hidden sentinel circuit, such that $q_{0}$ is the datum qubit and $q_{1}$ is the sentinel. It can be seen that the datum qubit, $q_{0}$, is set to H|0〉. The setting of the datum qubit is not important for the success of the circuit. The important characteristic is that the user did not read the the datum in the correct, namely the Hadamard, basis. This can be seen by the presence of an extra Hadamard gate on $q_{0}$ in the very middle of the Figure 10. This has the same meaning as in Figure 9. In Figure 10, there is the additional pair $q_{2}$ and $q_{3}$, with $q_{2}$ being the datum and $q_{3}$ the sentinel. A slight difference is that the datum has another initial setting, namely $q_{2}=|1\rangle$. As before, the initial value of $q_{2}=|1\rangle$ does not affect the capability of catching the intruder. As the intruder can be seen to erroneously read both $q_{1}$ and $q_{3}$, the chance to signal the presence of the intruder is increased. Theoretically, the signalling chance of the intruder in this case is $p=1-(\frac{3}{4}{)}^{2}\approx 0.44$. The measurements on the real computer *Belem* deviate from the expected theoretical result, as shown in Figure 11. On the positive side, the practical test shows that the intruder is caught with probability $p_{pr}=49.42$. Nevertheless, the problem is the unreliability; this number should not deviate that much from the theoretical expectation. In the cases where the server does not catch the intruder, namely, outcomes 0000,0001,0100,0101, the measured values deviate by 11.82%. In the cases where the server catches the intruder, which are all the others, the difference is even worse, namely 21.98%. The worrying situation shows at the very base state 0000, where the theoretical percentage should be 6.13%, but is actually 20.2%. It seems that the state of a qubit easily and spontaneously reverts back to the base state |0〉.

It remains to be seen that these ideas can be implemented once error free quantum computers are available.

#### 4.2.2. Sentinel Complexity Comparison

Positional sentinels versus hidden sentinels exhibit differences in terms of behavior, scope, implementation complexity, and cost.

The main difference is that positional sentinels are part of the data that are exposed to the intruder, whereas hidden sentinels are simply acted on by exposed data, but are not accessible to the user at all. As such, positional sentinels incur a simple cost of one qubit, while hidden sentinels need two qubits. Setting up a positional sentinel means simply setting the value of the qubit to a meaningful value. This means zero, one or two gates, depending on the value to be set. For hidden sentinels, additionally to setting the value of the datum, which is identical to the positional sentinel, the circuit requires two more Hadamard gates as well as two controlled phase shift gates. As controlled phase shift gates are two qubit gates, they are more complex and more prone to error. Thus, the hidden sentinels are more costly, both in terms of number of qubits as well as circuitry. The setting up of a hidden sentinel may also be considered more time consuming, though direct time evaluations of this action are hard to carry out for our limited experimental capacity. The below table offers a brief comparison. The table considers *N* sentinels.
Type ofNumber of qubitsNumber ofNumber ofSentinelfor *N* sentinelssingle qubit gatestwo qubit gatesPositional*N*from 00sentinel
to 2N
Hidden2Nfrom 2N2Nsentinel
to 4N
*The
resource comparison between positional and hidden sentinels**is done for N sentinels.*

Thus, hidden sentinels are more costly, but also more insidious.

The practicality of the methods presented here depends on the availability of quantum technology. The size of quantum computers today includes some tens of qubits. IBM quantum computers were based on 53 qubits in 2019. Some very rapid growth is expected in the near future. Quantum annealers have been reported as having 5000 qubits. Quantum networks, which allow quantum communication to happen, have stepped into the size of 700 optical fibers built in 2021 [17]. As this field is now growing on several fronts, the values given here may already be obsolete by the time the ink dries. The necessity of quantum honeypots may have to wait for a few more break-throughs in quantum technology.

## 5. Conclusions

This paper shows that a quantum network setting can contribute to the power of a honeypot system. This is because qubits can be checked for *reading* by adding sentinels to them. Sentinels can be added to as many qubits as wished for by the honeypot administrator. Quantum sentinels check whether a qubit, field, memory or disk location has been accessed for reading (only). There is no need for actual writing to detect the presence of illegal activity. Detecting the reading activity relies on quantum properties, such as the collapse of superposition and controlled quantum gates. Therefore, there is no possibility of mimicking the same capability by classical computational means.

Two types of quantum sentinels have been defined: positional-visible sentinels and hidden-invisible sentinels. The meaning captured in their name is that the are visible or invisible to the user of the honeypot system. In the case of hidden quantum sentinels, the illegal user is entirely unaware of the presence of the sentinel. The probability of catching an intruder on any sentinel is low and varies with the sentinel type. In the case of positional sentinels, the probability is $\frac{1}{4}$, and, in the case of hidden sentinels, the probability is $\frac{1}{8}$. Though hidden sentinels have the advantage of remaining hidden from the intruder, their drawback is a more complex quantum circuit with an extra qubit and a lower detection rate. Nevertheless, in both cases, the probability of catching an intruder can be increased to any arbitrary value by adding more sentinels.

Thus, quantum sentinels add the following properties to honeypots:The monitoring of malicious activity can be detailed to the level of bit, that is the information unit.The presence of the monitoring system can be fully hidden via hidden quantum sentinels.

Finally, today’s quantum computers do not offer the accuracy necessary to practically implement such quantum honeypots, as our experiments show. The errors of both signaling a legal user or ignoring an intruder deviate from the theoretical expectations by 10 to 20%. These values have been measured for two sentinels only. At this point, a larger experiment with more accurate error rates is not necessary, as even this value condemns the system as not being feasible as of yet. How soon this problem will be remedied remains an open question.

## Figures and Tables

**Figure 1 entropy-25-01461-f001:**
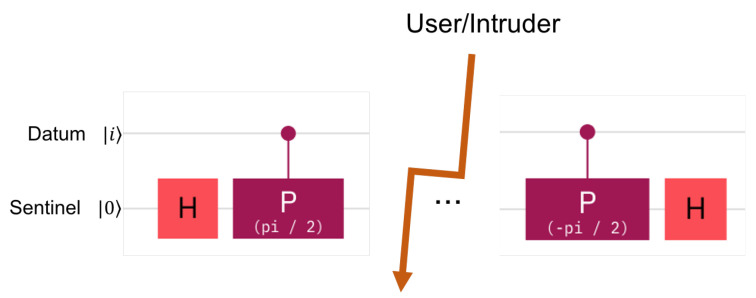
The hidden sentinel is the second qubit in the figure. It is acted on by the datum sentinel via the control of phase shift gates. The middle of the figure shows the area and time when the datum-qubit is exposed to the user.

**Figure 2 entropy-25-01461-f002:**
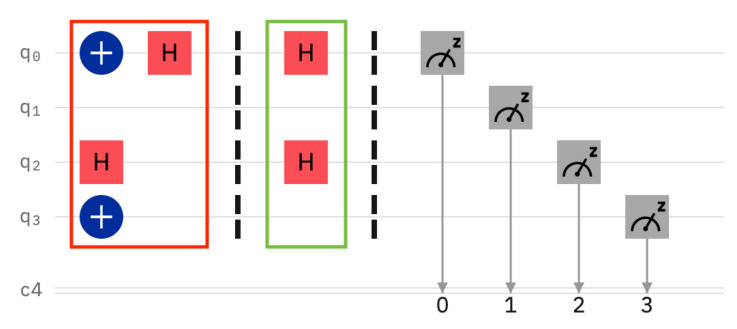
The legal user measures the sentinels in the correct bases.

**Figure 3 entropy-25-01461-f003:**
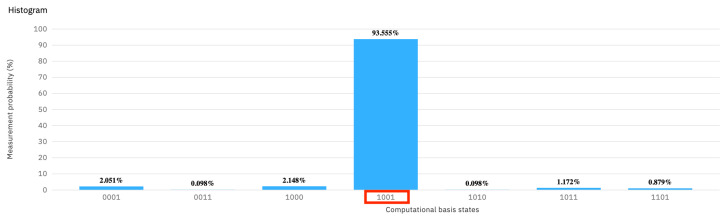
The measurement probability of Figure 2, which has four sentinels, and the user is legal.

**Figure 4 entropy-25-01461-f004:**
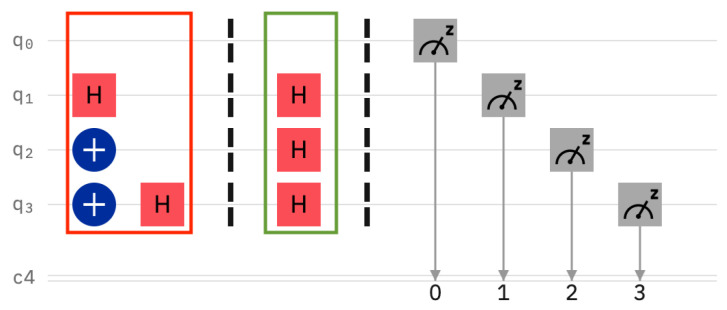
Experiment with four positional sentinels. The intruder’s behavior is random, as the state of the sentinels is not known to the user. In this particular case, the intruder makes a mistake on $q_{2}$ and, therefore, the detection probability is theoretically $\frac{1}{2}$.

**Figure 5 entropy-25-01461-f005:**
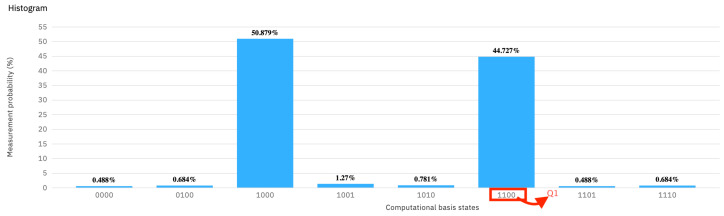
The measurement probability of Figure 4, which contains four positional sentinels, and the intruder misses one.

**Figure 6 entropy-25-01461-f006:**
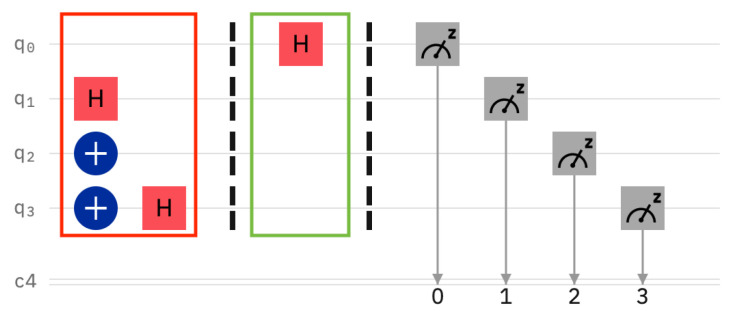
Four positional sentinels are set to all possible values. The experiment shows the option where the intruder is lucky on only one qubit, namely $q_{2}$.

**Figure 7 entropy-25-01461-f007:**
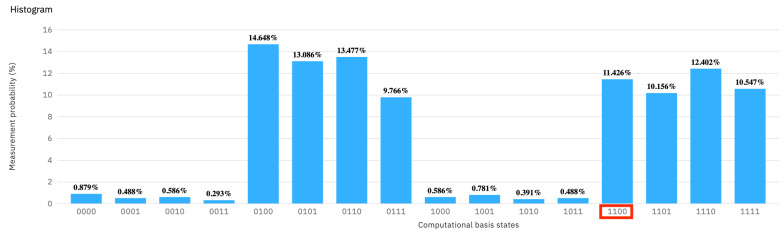
The measurement probability of Figure 6 with four positional sentinels and the intruder missing three of the sentinels. The result in red refers to the probability of the intruder to escape detection.

**Figure 8 entropy-25-01461-f008:**
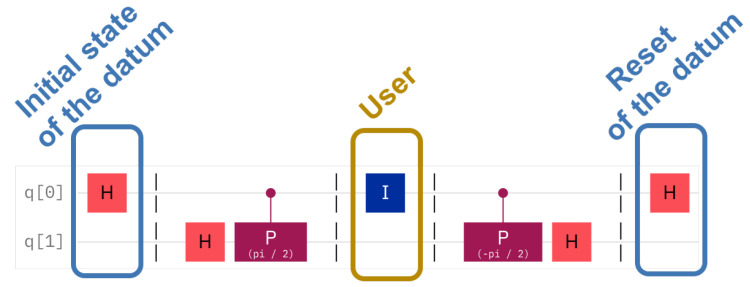
Legal user reading a datum qubit with a hidden sentinel. The same circuit applies to a lucky intruder. The user does not disturb the state of the datum qubit.

**Figure 9 entropy-25-01461-f009:**
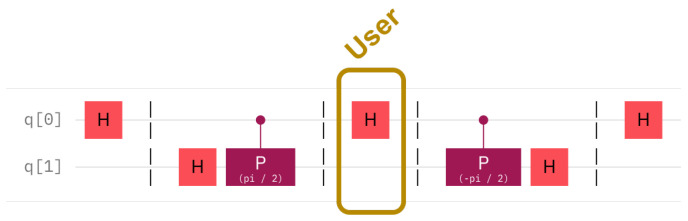
Unlucky intruder reading a quantum qubit with a hidden sentinel. In the case of an unlucky intruder, an extra Hadamard gate on the datum qubit disturbs the hidden sentinel.

**Figure 10 entropy-25-01461-f010:**
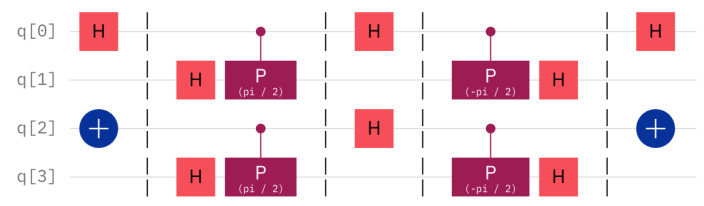
The circuit with two active hidden sentinels shows an intruder that has wrongly measured two datum qubits, $q_{0}$ and $q_{2}$, that act on two hidden sentinels, $q_{1}$ and $q_{3}$.

**Figure 11 entropy-25-01461-f011:**
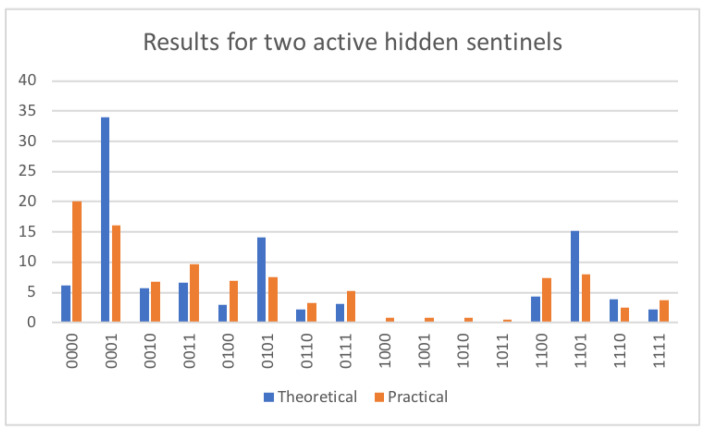
The left panel shows the practical results obtained on running a circuit with two hidden sentinels. The panel on the right shows the theoretical expectation.

**Table 1 entropy-25-01461-t001:** Positional sentinels and their behavior at reading.

Sentinel Value	Correct Reading Basis	User		Server	Probability to Catch
		Reading Type	Outcome	Measurement	
|0〉	Computational	honest, Computational	|0〉	|0〉	Not applicable
		intruder, Computational	|0〉	|0〉	25%
		intruder, Hadamard	|+〉 or |−〉	|0〉 or |1〉	
|1〉	Computational	honest, Computational	|1〉	|1〉	Not applicable
		intruder, Computational	|1〉	|1〉	25%
		intruder, Hadamard	|−〉 or |+〉	|1〉 or |0〉	
|+〉	Hadamard	honest, Hadamard	|+〉	|+〉	Not applicable
		intruder, Hadamard	|+〉	|+〉	25%
		intruder, Computation	|0〉 or |1〉	|+〉 or |−〉	
|−〉	Hadamard	honest, Hadamard	|−〉	|−〉	Not applicable
		intruder, Hadamard	|−〉	|−〉	25%
		intruder, Computation	|1〉 or |0〉	|−〉 or |+〉	

## Data Availability

No new data were created or analyzed in this study. Data sharing is not applicable to this article.

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
