# Peer review of "Quantum Honeypots"

_entropy, 2023, doi:10.3390/e25101461_

Round 1
Reviewer 1 Report
This manuscript describes a quantum honeypot system for intrusion detection in quantum computers. When intruders read certain quantum bits, such behavior could be detected based on the property of quantum bits, which could be measures in both computational or Hadamard base. Positional and hidden sentinels are developed to detect the intruders' existence. Since the intruder might use wrong bases to measure certain quantum bits, the quantum state might collapse to wrong base states. Then the intruder will be detected via reading quantum bits or entangled ones.
Since the quantum honeypot strategy is developed based on the property of quantum bits, its soundness is good. The writing is consistent.
Several issues should be addressed before publication:
1) For hidden sentinels, the reason to apply quantum Fourier transform should be elaborated with more details. Is it okay not to use it, but only based on the quantum entanglement?
2) The scenarios or when to use positional and hidden sentinels should be illustrated.
3) Multiple sentinels might increase the detection rate. The formal expression of the probability should be presented.
4) The circuit with two active hidden sentinels in Figure 10 needs further explanation.
5) Complexity comparison of positional and hidden sentinels as well as the possible delay should be analyzed with details.
6) Experiments could be improved. The current version is more like examples. The simulation of bigger systems might help demonstrate the scalability of the proposed scheme.
The writing is good. Sometimes, phrase "in the cases" is replaced by "on the cases". Further improvement is expected.
Author Response
Dear Reviewer, please look at the attached document, called "Reviewer1.docx"

Reviewer 2 Report
This study pioneers the concept of quantum honeypot for the detection of reading by adding quantum sentinels to the bit level.
The proposed idea is interesting and is of potential, yet severely limited by the error rate of quantum computers.
Some vision of the limitation should be addressed, rather leave it as an open question.
Author Response
We thank the reviewer for the time and effort spent on our paper as well as for the positive appreciation.
In answer to the comment
"Some vision of the limitation should be addressed, rather leave it as an open question."
we added a new \subsubsection called "Sentinel Complexity Comparison". The text is located on page 14, before the Conclusion.
We hope that with this addition, we have addressed the reviewer's concerns.